# High prevalence of *Strongyloides stercoralis* in people living with HIV: A critical health challenge in the Peruvian Amazon Basin

Silvia Otero-Rodriguez[1,2*], Martin Casapia-Morales[3,4,5], Viviana Pinedo-Cancino[5,6], Seyer Mego-Campos[6], Victoria-Ysabel Villacorta-Pezo[7], Jorge Parráguez-de-la-Cruz[8], Eva H. Clark[9], Esperanza Merino[1,2], Jose-Manuel Ramos-Rincon[2,10,11]

**1** Infectious Diseases Unit, Alicante General University Hospital, Alicante, Spain, **2** Alicante Institute of Sanitary and Biomedical Research (ISABIAL), Alicante, Spain, **3** Infectious Diseases and Tropical Medicine Service, Loreto Regional Hospital, Iquitos, Peru, **4** Medical Department, Asociación Civil Selva Amazónica, Iquitos, Peru, **5** Faculty of Human Medicine, National University of the Peruvian Amazon, Iquitos, Peru, **6** Laboratory for Research on Natural Antiparasitic Products of the Amazon (LIPNAA-CIRNA), National University of the Peruvian Amazon, Iquitos, Peru, **7** Clinical Laboratory, National University of the Peruvian Amazon, Iquitos, Peru, **8** Clinical Laboratory, Asociación Civil Selva Amazónica, Iquitos, Peru, **9** Department of Medicine (Infectious Diseases) and Department of Pediatrics (Tropical Medicine), Baylor College of Medicine, Houston, Texas, United States of America, **10** Internal Medicine Department, Alicante General University Hospital, Alicante, Spain, **11** Clinical Medicine Department, Miguel Hernández University of Elche, Elche, Spain

* o.silvia.r@gmail.com

## Abstract

### Introduction

Strongyloidiasis is an important but underdiagnosed soil-transmitted helminthiasis, particularly in tropical areas and some vulnerable groups.

### Objectives

To assess the parasitological prevalence, seroprevalence and sociodemographic factors of *Strongyloides stercoralis* infection in patients living with human immunodeficiency virus (PLWH) in an endemic area.

### Materials and methods

We performed a cross-sectional study of strongyloidiasis in 537 PLWH in two hospitals in Iquitos, Peru, from 20 Oct 2023 to 20 May 2024. We tested patient sera using Strongyloides IgG enzyme-linked immunosorbent assay (ELISA) and stool via the modified Baermann technique and/or charcoal fecal culture as highly sensitive parasitological techniques. We used multivariable logistic regression to identify factors associated with *S. stercoralis* infection.

**Data availability statement:** The dataset used and/or analysed during the current study are available in Zenodo Repository, under the ORCID: 10.5281/zenodo.14864472. The laboratory protocol is available in protocols.io, under the doi: https://dx.doi.org/10.17504/protocols.io.ewov1mpqpvr2/v1

**Funding:** This work was supported by Miguel Hernandez University (UMH) (under Grant 11-134-4-2023-0133 to J.-M.R), Alicante Health and Biomedical Research Institute (ISABIAL) (under Grant 2024-0181 to S.O.-R) and Instituto de Salud Carlos III (ISCIII) (under Grant CM23/00050 to S.O.-R). The funders had no role in study design, data collection and analysis, decision to publish, or preparation of the manuscript.

**Competing interests:** I have read the journal's policy and the authors of this manuscript have the following competing interests: Miguel Hernandez University has granted the research of J.M.-R in the field. Alicante Health and Biomedical Research Institute (ISABIAL) and Instituto de Salud Carlos III (ISCIII) have granted the research of S.O.-R in the field. None of the institutions had a role in study design, data collection, analysis, decision to publish, or preparation of the manuscript. The other authors declare that no competing interests exist.

## Results

Among the 339 PLWH whose stool samples were collected, 82 were positive for *S. stercoralis* (prevalence 24.2%; 95% confidence interval [CI] 20.0-29.1%). Among the 534 PLWH whose serum samples were collected, 227 were positive (seroprevalence: 42.5%; 95% CI 38.1-47.5%). The kappa value for charcoal culture and Baermann technique was 0.69. ELISA showed a sensitivity of 92.6% and a negative predictive value of 96.9%. Significant risk factors for stool positivity included living in a rural (unpaved) area (adjusted OR: 1.86), whereas significant risk factors for both stool and seropositivity included living in a poor house (made of wood/leaves) (adjusted odds ratio (ORs): 2.18 and 2.48, respectively), in the Loreto Regional Hospital catchment area (adjusted ORs: 5.66 and 5.37, respectively), or being infected by hookworms in stool (adjusted ORs: 23.88 and 9.78, respectively). Having a low level of studies was associated with seropositivity (adjusted OR 2.42).

## Conclusion

The prevalence of *S. stercoralis* is high among PLWH in Iquitos, especially among those living in conditions of socioeconomic vulnerability or co-infected with hookworms. The negative predictive value of the *S. stercoralis* ELISA was high, although this result should be taken with caution in severe immunosuppression.

## Author summary

Intestinal parasitic infections are especially common in warm climates, particularly where sanitation facilities are poor. Immunocompromised individuals, such as people living with human immunodeficiency virus (HIV) patients, are more susceptible to parasitic infections. *Strongyloides stercoralis* is a common parasite in Iquitos, the capital of the Peruvian Amazon, although its current infection rate among HIV patients is unknown. In this study, our objective was to detect the prevalence of the parasite in the stool of 339 patients from two different hospitals using three microbiological techniques, as none of them is completely reliable by itself. The prevalence of antibodies in the blood of 537 patients was also assessed, since it is useful for knowing the burden of disease in the community as a contact marker. In addition to the diagnostic procedure, we conducted an interview on sociodemographic and clinical characteristics to identify risk factors for infection that can be corrected. This information will help improve prevention, diagnosis, and clinical management in this population.

## 1. Introduction

Strongyloidiasis is a soil-transmitted helminthiasis caused by the parasitic nematode *Strongyloides stercoralis* (distributed worldwide) or *Strongyloides fuelleborni* (only

present in some regions of Africa and Oceania) [1]. This infection is particularly prevalent in tropical and subtropical areas, where sanitary conditions are poor, and t is acquired through skin contact with contaminated soil [2]. The global prevalence of *S. stercoralis* is estimated to affect approximately 600 million people [3,4], although this figure may be underestimated due to the low sensitivity of the available diagnostic techniques. Strongyloidiasis is important because it is one of the few nematodes able to perpetuate a chronic infection in the host ("autoinfection"). Without treatment, the infection can persist for life and immunosuppressed patients are at risk for *Strongyloides* hyperinfection syndrome and disseminated strongyloidiasis, conditions that carry high morbidity and can even be fatal [5]. The World Health Organization (WHO) includes *Strongyloides* among the soil transmitted helminthiases targeted for improved control by 2030; for this purpose, specific guidelines have been published recently. However, the organization denounces the need for more evidence to optimize public health programs for strongyloidiasis [6].

Iquitos is a city located in the Loreto region, which has the second-highest prevalence of people living with HIV in Peru, with more than 200 new cases diagnosed in the first half of 2024 [7]. However, the prevalence of *S. stercoralis* infection in PLWH in Peru is understudied, as their predisposition to infection, the impact of HIV related immunosuppression, and the associated risk of hyperinfection syndrome. Despite this, we do know that this is a vulnerable population with a higher risk for high-dose corticosteroid use (a known risk factor for hyperinfection [8]), and in which treating the parasite would reduce morbimortality.

Population studies of *S. stercoralis* infection in Peru show widely varying prevalences depending on the geographical area, the type of diagnostic test used, or the clinical presentation. A study conducted by the Peruvian Ministry of Health, from 1981 to 2010, found a heterogenous prevalence ranging from 0.3 to 39%, with a national average of 6.25% [9]. A recent systematic review estimated an even higher prevalence of 7.34% [10]. In the Peruvian Amazon, where climatic conditions favor the endemicity of the parasite, the prevalence of *S. stercoralis* consistently exceeds 10% [11–13]. To the best of our knowledge, no published parasitological or sero-epidemiological studies exist on *S. stercoralis* infection in PLWH in the Peruvian Amazon.

The parasitological diagnosis of *S. stercoralis* infection is challenging. Stool processing techniques including the Baermann method and charcoal culture are diagnostic tools that yield higher sensitivity than direct stool smear, but still miss a large proportion of infections [5]. Molecular methods like PCR allow the diagnosis of a higher number of cases; however, this procedure is not available in most patient care settings. While serologic tests are the most sensitive diagnostic tools (especially in non-endemic areas), false positive results can occur due to cross-reactions with other helminth infections and the long-term persistence of *Strongyloides* antibodies even after treatment [14]. Overall, only a combination of techniques can be accepted as highly sensitive for the diagnosis of strongyloidiasis [4].

This study assessed the parasitological prevalence, seroprevalence, and risk factors for *S. stercoralis* infection in PLWH in an endemic area. Our results highlight the urgent need for improved diagnostic tools and expanded public health policies to improve strongyloidiasis diagnosis and treatment strategies for this vulnerable population.

## 2. Methods

### 2.1. Ethics statement

The Ethics Committee of Loreto Regional Hospital in Iquitos (Peru) (EXP: ID-018-CIEI-2013) approved this study. After being informed about the study, individuals who volunteered to participate provided written consent to be included. We maintained all the results in strict confidentiality, and those who tested positive for intestinal parasites received free treatment and follow-up by their HIV healthcare provider.

### 2.2. Study population and inclusion/exclusion criteria

This was an observational, cross-sectional study of PLWH receiving care at one of two hospitals in Iquitos, Peru: (1) the Regional Hospital of Loreto "Felipe Santiago Arriola Iglesias" (which follows patients from the districts of Punchana and Iquitos, but also from the rest of the Loreto Health Department), and (2) the Hospital of Iquitos "César Garayar García" (which follows patients from the districts of Belen and San Juan), from 20 Oct 2023 to 20 May 2024.

We included patients over 18 years with known HIV infection, attending the Regional Hospital of Loreto (located in Iquitos' Punchana district) or the Hospital of Iquitos (located in Iquitos' San Juan district) (Fig 1), who were able to provide stool and/or blood specimens. We offered enrollment to both eligible outpatients and inpatients.

## 2.3. Enrollment procedures

After providing informed consent, study participants provided a stool specimen for *S. stercoralis* larvae copro-parasitological examination (given the complexity of collecting it at home, only one sample per patient was required), and a blood specimen for *S. stercoralis* serology. We collected epidemiological demographic and risk factor variables using an oral semi-structured interview and clinical history of the patient, if available.

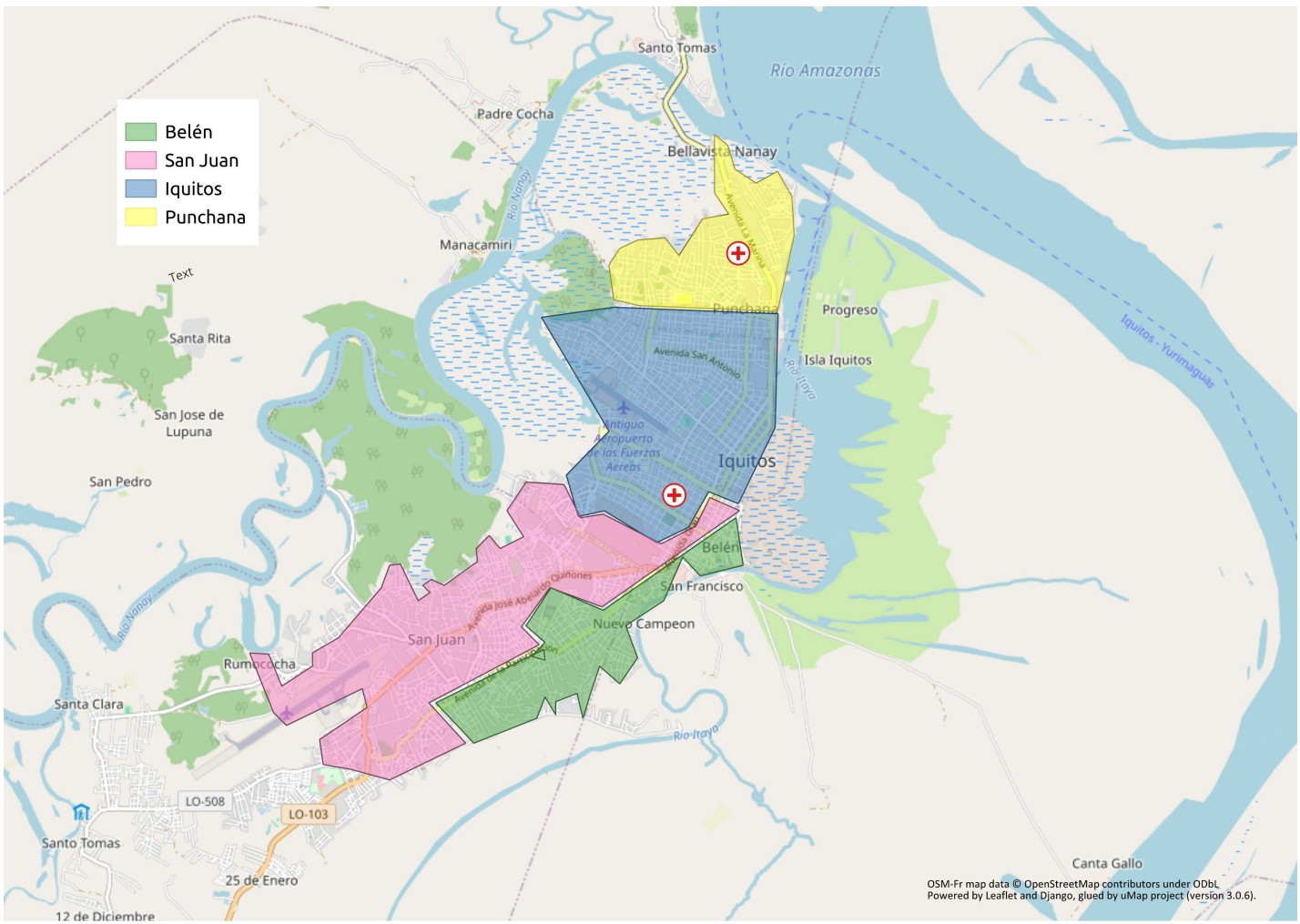

**Fig 1.  Map of Iquitos (Peru), with its four major districts.** From north to south: Punchana district, Iquitos district, San Juan Bautista district and Belen district. The red crosses symbolized the two hospitals included in the study (from north to south: Loreto Regional Hospital and Iquitos Hospital). *Map created with uMap project (version 3.0.6), using data from OpenStreetMap and OpenStreetMap Foundation, licensed under ODbL.* https://www.open-streetmap.org/#map=12/-3.7432/-73.2342. *Custom data and layers included in this map are licensed under Creative Commons BY-SA 4.0.* https://www.openstreetmap.org/copyright/. https://creativecommons.org/licenses/by-sa/4.0/.

## 2.4. Stool examination for *S. stercoralis*

Each fecal specimen was analyzed using three techniques: direct examination with Lugol's iodine, modified Baermann technique, and charcoal fecal culture. A specimen was considered positive when *S. stercoralis* larvae were identified by any of these techniques.

- Modified Baermann technique [15]. Briefly, five grams of fresh feces were placed in the center of a cotton-wool gauze sieve, positioned in a funnel partially submerged in a sedimentation flask filled with water at 37 °C. After one hour at room temperature (25-37ºC), larvae migrated from the fecal suspension into the heated water. The supernatant was discarded, and 1 mL of sediment from the funnel bottom was microscopically examined for the presence of larvae.

- Charcoal culture (Dancescu culture) [16]: Briefly, four grams of fresh feces were mixed with equal parts of distilled water and granulated charcoal. The mixture was placed in a Petri dish, sealed with vinyl tape, and incubated at 30 °C in darkness. The culture was examined with a compound microscope for *S. stercoralis* adult worm (free-living or filariform) on the second, fourth, and seventh days before discarding the culture.

## 2.5. *S. stercoralis* serology

Blood samples were centrifuged at 2058 relative centrifugal force for 10 minutes to separate plasma from erythrocytes. The plasma was stored at -80°C until analysis. For serological testing, we employed a commercially available *Strongyloides* crude antigen IgG enzyme-linked immunosorbent assay (ELISA), specifically the *Strongyloides* IgG IVD-ELISA kit (DRG Instruments GmbH, Marburg, 278 Germany, approved by the European Commission). This kit utilizes microtiter wells coated with a soluble fraction of the *S. stercoralis* L3 filariform larval antigen. All assays were performed following the manufacturer's protocol. Considering the anticipated high seroprevalence and potential false positives near the manufacturer's recommended cut-off value (0.200), we conducted duplicate ELISA testing for the initial 100 participants. Based on these preliminary results, we established an optimized cut-off value of 0.220 for determining test positivity.

## 2.6. Data analysis

Statistical analyses were performed via IBM SPSS Statistics version SPSS 22.0 (IBM, Armonk, EEUU). For descriptive statistics, categorical variables were expressed as frequencies and percentages, while continuous variables were presented as medians with interquartile range (IQR). The 95% confidence intervals were calculated using the Newcombe method [17]. Categorical variables were compared using Chi-square tests, while continuous variables were analyzed using the U-Man Whitney test. Agreement between modified Baermann and charcoal culture results was assessed using Cohen's Kappa coefficient.

Risk factors for *S. stercoralis* infection were initially evaluated through bivariate analysis, with associations quantified using ORs. Subsequently, multivariable logistic regression models were constructed to identify independent risk factors for both parasitological *S. stercoralis* infection and seropositivity to *S. stercoralis*. These models included age and sex, and variables that showed statistical significance ($p \leq 0.05$) in the univariate analyses. The models' goodness of fit was assessed using CoxSnell $R^2$ and Nagelkerke $R^2$ statistics to determine the strength of association between dependent variables (parasitological infection *S. stercoralis* infection and seropositivity to *S. stercoralis*) and independent variables.

We calculated the sensitivity, specificity, positive predictive value and negative predictive value for the ELISA results, along with their respective 95% CIs, using the parasitologic results (i.e., outcomes from the modified Baermann technique and/or charcoal culture) as the reference standard. Same analysis was realized also including other helminths infections, in other to rule out cross-reactions.

# 3. Results

## 3.1. Description of the study population and epidemiologic data

For the 537 PLWH included in this study, we obtained serum from 534 and stool from 339 patients (Fig 2). More than 60% were heterosexual men with few comorbidities, and the median age was 41 years (range 32-49) (Table 1). Most patients had well-controlled HIV with an undetectable HIV viral load.

## 3.2. *S. stercoralis* stool examination (parasitologic) results

Among the 339 participants with available stool samples, only 1 was positive in the direct exam, while 82 tested positive by the modified Baermann technique and/or charcoal culture (prevalence 24.2%; 95% CI 20.0-29.1%). Five were positive via the Baermann technique alone, 30 via charcoal culture alone, and 47 via both tests. The Kappa index was 0.69 (95% CI: 0.58-0.79), indicating a good correlation between the modified Baermann method and charcoal culture (Table 2).

## 3.3. Other Helminths isolated in stool examination

Among the 339 patients with stool specimen, 17 (5.0%) were positive for *hookworms*, 10 (2.9%) for *Ascaris lumbricoides*, 2 (0.6%) for *Hymenolepis nana* and 1 (0.3%) for *Trichuris trichiura.*

Among the 82 patients who were positive for *S. stercoralis* infection in stool, 15 (18.2%) were also co-infected by hookworms, 6 (7.3%) coinfected by *Ascaris lumbricoides* and 1 (1%) co-infected by *Trichuris trichiura.*

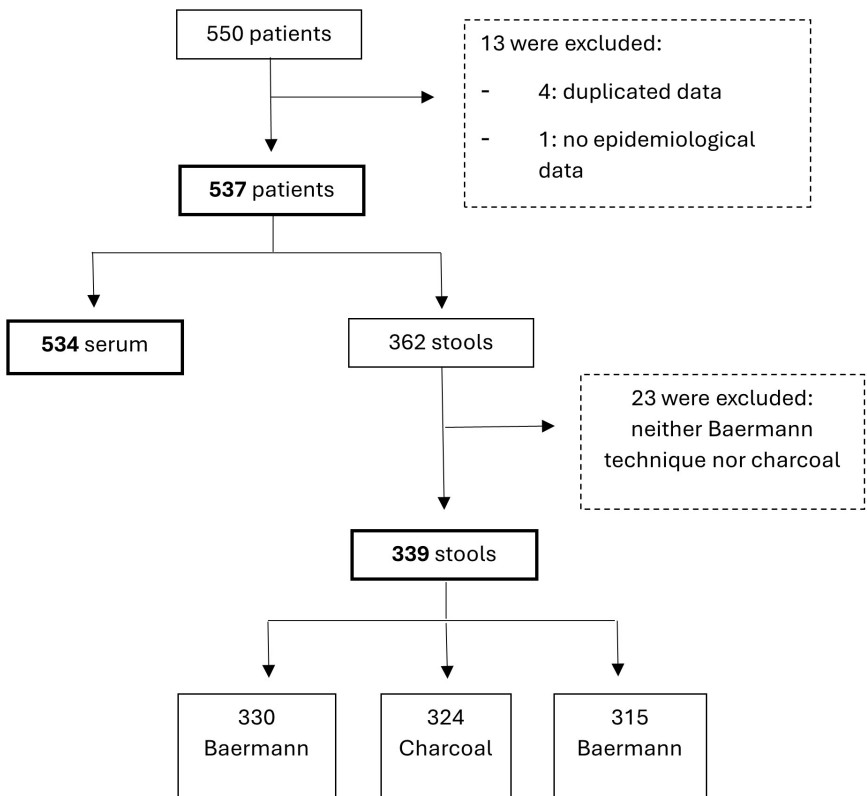

**Fig 2. Flow chart of study participant inclusion and specimen availability for the study.**

**Table 1. Epidemiological characteristics of study participants, divided by those with available serum specimens and those with stool specimens.**

| Variables | Patients with serum specimen (N = 534) | Patients with stool specimen (N = 339) |
|---|---|---|
| **Male**, % (n/N) | 66.1 (353/534) | 64.3 (218/339) |
| **Age**, median (IQR), years | 41 (32, 49) | 41 (32, 48) |
| **Hospital attended**, % (n/N) | | |
| Regional Hospital of Loreto | 77.9 (416/534) | 84.7 (287/339) |
| Inpatient | 1.5 (8/534) | 0.3 (1/339) |
| Outpatient | 76.4 (408/534) | 84.4 (286/339) |
| Hospital of Iquitos (Outpatient) | 22.1 (118/534) | 15.3 (52/339) |
| **Residence**, % (n/N) | | |
| Iquitos district | 32.0 (171/534) | 35.1 (119/339) |
| Punchana district | 25.1 (134/534) | 25.7 (87/339) |
| San Juan district | 20.4 (109/534) | 20.4 (69/339) |
| Belén district | 16.3 (87/534) | 14.7 (50/339) |
| Out of Iquitos metropolitan area | 6.1 (33/534) | 4.1 (14/339) |
| **Occupation**, % (n/N) | | |
| Unemployed | 40.3 (215/534) | 43.4 (147/339) |
| Primary Sector | 18.2 (97/534) | 14.5 (49/339) |
| Secondary sector | 33.1 (177/534) | 4.4 (15/339) |
| Tertiary sector | 8.4 (45/534) | 10.0 (34/339) |
| **Education**, % (n/N) | | |
| None | 2.2 (12/534) | 2.4 (8/339) |
| Completed primary school | 18.9 (101/534) | 17.4 (59/339) |
| Completed secondary school | 54.3 (290/534) | 55.8 (189/339) |
| Attended university | 24.5 (131/534) | 24.5 (83/339) |
| **Epidemiological risk factors**, % (n/N) | | |
| Lives with dogs, cats or farm animals | 69.1 (369/534) | 69.9 (237/339) |
| Walks barefoot | 26.8 (143/534) | 26.8 (91/339) |
| Resid in rural location[a] | 33.1 (177/534) | 31.0 (105/339) |
| Lives in house made of wood or leaves | 48.0 (256/534) | 47.2 (160/339) |
| Alcohol or tobacco consumption | 51.7 (276/234) | 52.5 (178/339) |
| Albendazole[b] 6 months prior to study | 8.8 (47/534) | 5.0 (17/339) |
| Pregnancy, % (n/N) | 0.9 (5/534) | 0.9 (3/339) |
| **Comorbidity**, % (n/N) | | |
| Diabetes or high blood pressure | 6.7 (36/534) | 7.4 (25/339) |
| Other cardiovascular disease | 2.8 (15/534) | 3.2 (11/339) |
| Digestive disease | 7.1 (38/534) | 5.9 (20/339) |
| Urinary disease | 2.1 (11/534) | 2.1 (7/339) |
| Dermatological disease | 0.6 (3/534) | 0.6 (2/339) |
| Other | 1.1 (6/534) | 0.3 (1/339) |
| **Previous infections**, % (n/N) | | |
| Intestinal parasitosis | 18.2 (97/534) | 13.3 (45/339) |
| Chronic hepatitis | 6.7 (36/534) | 6.8 (23/339) |
| Gonorrhea | 11.6 (62/534) | 13.3 (45/339) |
| Syphilis | 13.5 (72/534) | 14.2 (48/339) |
| Tuberculosis | 23.2 (124/534) | 21.8 (74/339) |

*(Continued)*

**Table 1.** (Continued)

| Variables | Patients with serum specimen (N = 534) | Patients with stool specimen (N = 339) |
|---|---|---|
| Cerebral toxoplasmosis | 3.2 (17/534) | 3.5 (12/339) |
| **Symptoms**, % (n/N) | | |
| Cough or cold symptoms | 14.8 (80/534) | 13.8 (74/339) |
| Fever | 1.7 (9/534) | 1.2 (4/339) |
| Diarrhea | 20.4 (109/534) | 21.8 (74/339) |
| < 4 times a month | 79.8 (87/109) | 77.0 (57/74) |
| >= 4 times a month | 20.2 (22/109) | 23.0 (17/74) |
| **Risk group**, % (n/N) | | |
| Heterosexual | 75.4 (376/499) | 76.5 (244/319) |
| Homosexual | 18.2 (97/499) | 18.8 (60/319) |
| Transexual | 2.1 (11/499) | 2.5 (8/319) |
| Bisexual | 3.0 (15/499) | 2.2 (7/319) |
| **HIV acquisition**, % (n/N) | | |
| Sexual | 87.3 (466/534) | 90.6 (307/339) |
| Vertical | 0.7 (4/534) | 0.3 (1/339) |
| Parenteral | 0.2 (1/534) | 0.3 (1/339) |
| Unknown | 11.8 (63/534) | 8.8 (30/339) |
| **CD4 + nadir**, median (IQR),/ml | 238 (118, 375) | 239 (120, 375) |
| **Current CD4 +,** median (IQR),/ml | 443 (281, 615) | 432 (289, 587) |
| **Current CD4+ < 200/ml** (AIDS), % (n/N)/ml | 13.9 (52/374) | 12.2 (29/238) |
| **Current HIV viral load**, % (n/N) (copies/ml) | | |
| Undetectable < 20 | 75.8 (380/501) | 76.9 (250/325) |
| Between 20 and 2000 | 11.4 (61/501) | 11.4 (37/325) |
| Detectable > 2000 | 11.2 (60/501) | 11.7 (38/325) |
| **ART adherence**, % (n/N) | | |
| >95% | 85.6 (374/437) | 85.9 (243/283) |
| 90-95 | 6.9 (30/437) | 6.4 (18/283) |
| <90 | 7.6 (33/437) | 7.8 (22/283) |

Data shown as % (n/N) or median (IQR), unless specified otherwise. Statistically significant differences shown in bold. Percentages may not total 100 due to rounding. [a]Defined as absence of paved streets. [b]Self-medication with Albendazole (dose unknown). Abbreviations AIDS: Acquired Immune Deficiency Syndrome. ART: anti-retroviral therapy.

**Table 2.** Correlation between charcoal culture and modified Baermann technique.

| | | Charcoal culture | | Total |
|---|---|---|---|---|
| | | Positive, n (%) | Negative, n (%) | |
| **Modified Baermann technique** | Positive, n (%) | 47 (14.9%) | 2 (0.6%) | 49 (15.6%) |
| | Negative, n (%) | 30 (9.5%) | 236 (74.9%) | 266 (84.4%) |
| **Total** | | 77 (24.4%) | 238 (75.6%) | **315 (100%)** |

### 3.4. *S. stercoralis* serologic results

Among the 534 participants with available serum specimens, 227 tested positive by ELISA (seroprevalence: 42.5%; 95% CI 38.1-47.5%).

### 3.5. Risk factors associated with parasitological *S. stercoralis* infection

After adjusting for sex, age, and variables with p values ≤ 0.05 in the bivariate analysis (Table 3), the characteristics most strongly associated with parasitological infection of *S. stercoralis* in PLWH were belonging to the Loreto Regional Hospital catchment area (adjusted OR: 5.43), living in a rural area (adjusted OR: 1.86), living in a house made of wood/leaves (adjusted OR: 2.18) and having hookworms in stools (adjusted OR: 23.88) (Fig 3a). The model's Cox–Snell $R^2$ value was 0.18 and the Nagelkerle $R^2$ value was 0.27, with an AUC of 0.76 (95% CI 0.70 – 0.82, $p < 0.001$).

**Table 3. Variables associated with *S. stercoralis* infection defined by visualization of larvae in stool (via the modified Baermann technique or charcoal culture).**

| | Infection defined by larvae in stool (N = 339) | | | |
| --- | --- | --- | --- | --- |
| | Infected (N = 82) | Not Infected (N = 257) | OR[c] (IC95%) | p |
| **Sex** (male), % (n/N) | 68.3 (56/82) | 63.0 (162/257) | 1.26 (0.74-2.15) | 0.387 |
| **Age** (years), median (IQR) | 41 (31, 48) | 41 (32, 49) | – | 0.746 |
| **Hospital attended**, % (n/N) | | | | |
| Hospital Regional de Loreto | 95.1 (78/82) | 81.3 (209/257) | 4.48 (1.56-12.83) | **0.003** |
| Hospital de Iquitos | 4.9 (4/82) | 18.7 (48/257) | 1 | |
| **Residence** (Punchana district), % (n/N) | 28.0 (23/82) | 24.9 (64/257) | 1.18 (0.67-2.06) | 0.570 |
| **Occupation** (primary sector), % (n/N) | 17.0 (14/82) | 13.6 (35/257) | 1.31 (0.66-2.57) | 0.439 |
| **Education** (illiterate or completed only primary school), % (n/N) | 24.4 (20/82 | 18.3 (47/257) | 1.44 (0.80-2.61) | 0.227 |
| **Epidemiological risk factors**, % (n/N) | | | | |
| Lives with dogs, cats or farm animals (yes) | 63.4 (52/82) | 72.0 (185/257) | 0.68 (0.40-1.14) | 0.141 |
| Walks barefoot (yes) | 28.0 (23/82) | 26.5 (68/257) | 1.08 (0.62-1.89) | 0.777 |
| Resid in rural location[a] (yes) | 45.1 (37/82) | 26.5 (68/257) | 2.29 (1.36-3.83) | **0.001** |
| Lives in house made of wood or leaves (yes) | 64.6 (53/82) | 41.6 (107/257) | 2.56 (1.53-4.29) | **<0.001** |
| Alcohol or tobacco consumption (yes) | 54. (45/82) | 51.8 (133/257) | 1.13 (0.69-1.87) | 0.621 |
| Albendazole[b] 6 months prior to study (yes) | 0.0 (0/82) | 6.6 (17/257) | – | **0.016** |
| **Previous infections**, % (n/N) | | | | |
| Chronic hepatitis (yes) | 8.5 (7/82) | 6.2 (16/257) | 1.41 (0.56-3.55) | 0.469 |
| STI (gonorrhea or syphilis) (yes) | 22.0 (18/82) | 21.0 (54/257) | 1.06 (0.58-1.93) | 0.856 |
| Tuberculosis (yes) | 18.3 (15/82) | 23.0 (59/257 | 0.75 (0.40-1.41) | 0.373 |
| Cough, fever or diarrhea (yes), % (n/N) | 12.2 (10/82) | 14.8 (38/257) | 0.80 (0.38-1.69) | 0.558 |
| Frequent diarrhea (>= 4 times/month), % (n/N) | 3.7 (3/82) | 5.4 (14/257) | 0.66 (0.19-2.35) | 0.772 |
| Risk group (non-heterosexual), % (n/N) | 23.2 (19/82) | 21.8 (56/257) | 1.14 (0.63-2.08) | 0.670 |
| **Current CD4+ < 200** (AIDS), % (n/N) | 8.5 (7/82) | 8.6 (22/257) | 0.96 (0.39-2.38) | 0.931 |
| **Current HIV viral load > 2000** (detectable), % (n/N) | 14.6 (12/82) | 11.3 (29/257) | 1.44 (0.70-2.97) | 0.324 |
| **ART adherence <= 95%** (bad adherence), % (n/N) | 12.2 (10/82) | 11.7 (30/257) | 1.09 (0.50-2.36) | 0.832 |
| **Hookworms in stool** | 18.3 (15/82) | 0.8 (2/257) | 28.55 (6.37-127.90) | **<0.001** |

Data shown as % (n/N) or median (IQR), unless specified otherwise. Statistically significant differences shown in bold. Percentages may not total 100 due to rounding. [a]Defined as absence of paved streets. [b]Self-medication with Albendazole (dose unknown). [c]Odds Ratio: "Not Infected" is the reference category. STI: sexually transmitted infection.

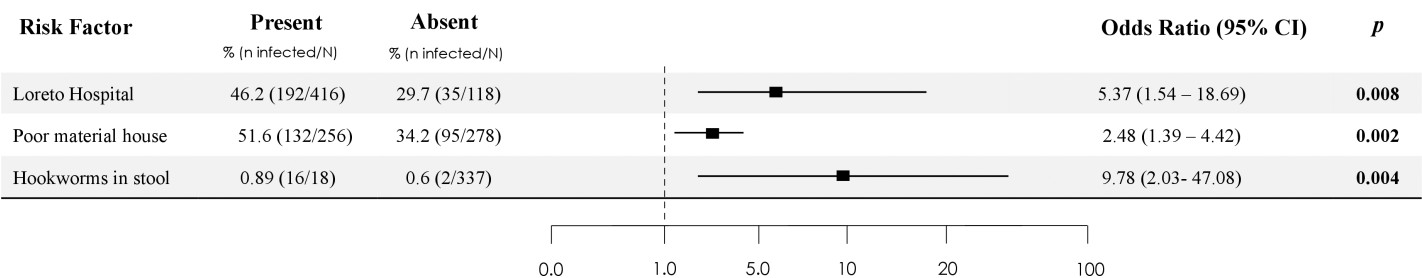

**a.** Independent Predictors of *S. stercoralis* infection (defined by larvae in stools) from Multivariable Logistic-Regression Analysis

| Risk Factor | Present<br>% (n infected/N) | Absent<br>% (n infected/N) | | Odds Ratio (95% CI) | *p* |
|---|---|---|---|---|---|
| Loreto Hospital | 27.1 (78/287) | 7.7 (4/52) | | 5.66 (1.88 – 17.05) | **0.002** |
| Rural area | 35.2 (37/105) | 19.2 (45/234) | | 2.16 (1.19 – 3.90) | **0.011** |
| Poor material house | 33.1 (53/160) | 16.2 (29/179) | | 2.10 (1.17 - 3.75) | **0.012** |
| Hookwoms in stool | 88.2 (15/17) | 0.6 (2/322) | | 23.88 (5.00 - 114.18) | **<0.001** |

**b.** Independent Predictors of *S. stercoralis* seropositivity (defined by positive ELISA in serum) from Multivariable Logistic-Regression Analysis

| Risk Factor | Present<br>% (n infected/N) | Absent<br>% (n infected/N) | | Odds Ratio (95% CI) | *p* |
|---|---|---|---|---|---|
| Loreto Hospital | 46.2 (192/416) | 29.7 (35/118) | | 5.37 (1.54 – 18.69) | **0.008** |
| Poor material house | 51.6 (132/256) | 34.2 (95/278) | | 2.48 (1.39 – 4.42) | **0.002** |
| Hookworms in stool | 0.89 (16/18) | 0.6 (2/337) | | 9.78 (2.03- 47.08) | **0.004** |

**Fig 3. a. Independent Predictors of *S. stercoralis* infection (defined by larvae in stools) from Multivariable Logistic-Regression Analysis. b. Independent Predictors of *S. stercoralis* seropositivity (defined by positive ELISA in serum) from Multivariable Logistic-Regression Analysis.**

### 3.6. Risk factors associated with *S. stercoralis* seropositivity

After adjusting by sex, age, and variables with p values ≤ 0.05 in the bivariate analysis (Table 4), four variables had a significant association with *S. stercoralis* seropositivity: belonging to Loreto Regional Hospital catchment area (adjusted OR: 3.88), living in a house made of wood/leaves (adjusted OR: 2.82), having hookworms in stools (adjusted OR: 9.78), and having a low level of education (illiteracy or primary school) (adjusted OR 2.42) (Fig 3b). The model's Cox–Snell $R^2$ value was 0.17 and the Nagelkerle $R^2$ value was 0.23, with an AUC of 0.72 (95% CI 0.67 – 0.78, $p < 0.001$).

### 3.7. Detection of *S. stercoralis* by serology versus stool examination

A total of 354 participants had both stool and serologic results. With the modified Baermann technique and/or charcoal culture as the parasitological reference standard, ELISA had a sensitivity of 92.6% and a negative predictive value of 96.9% (Table 5).

Only one hookworm infection was found to have a positive serology in absence of *S. stercoralis* stool infection.

## Discussion

We detected a high prevalence of *S. stercoralis* infections in this population; approximately 1 in 4 PLWH had *S. stercoralis* larvae identified in stool and the seroprevalence was higher than 40%. These findings underscore the importance of developing public health campaigns to screen populations living in tropical regions for *S. stercoralis* infection. Published

**Table 4. Variables associated with positive serology against *S. stercoralis*.**

| | Infection defined by positive ELISA (N = 534) | | | |
| --- | --- | --- | --- | --- |
| | Infection (N = 227) | Non-Infection (N = 307) | OR[c] (IC95%) | p |
| **Sex** (male), % (n/N) | 71.4 (162/227) | 62.2 (191/307) | 1.51 (1.05-2.19) | **0.027** |
| **Age** (years), median (IQR) | 41 (31, 49) | 41 (33, 48) | – | 0.942 |
| **Health area**, % (n/N) | | | | |
| Hospital Regional de Loreto | 84.6 (192/227) | 73.0 (224/307) | 2.03 (1.31-3.16) | **0.001** |
| Hospital de Iquitos | 15.4 (35/227) | 27.0 (83/307) | 1 | |
| **Residence** (Punchana district), % (n/N) | 27.8 (63/227) | 23.1 (71/307) | 1.28 (0.86-1.89) | 0.223 |
| **Occupation** (primary sector), % (n/N) | 21.6 (49/227) | 15.6 (48/307) | 1.49 (0.96-2.31) | 0.078 |
| **Education** (illiterate or completed only primary school), % (n/N) | 25.1 (57/227) | 18.2 (56/307) | 1.50 (1.00-2.28) | **0.050** |
| **Epidemiological risk factors**, % (n/N) | | | | |
| Lives with dogs, cats or farm animals (yes) | 69.2 (157/227) | 69.1 (212/307) | 1.01 (0.69-1.46) | 0.979 |
| Walks barefoot (yes) | 27.8 (63/227) | 26.1 (80/307) | 1.09 (0.74-1.60) | 0.662 |
| Resid in rural location[a] (yes) | 36.1 (82/227) | 30.9 (95/307) | 1.26 (0.88-1.82) | 0.209 |
| Lives in house made of wood or leaves (yes) | 58.1 (132/227) | 40.4 (124/307) | 2.05 (1.45-2.91) | **<0.001** |
| Alcohol or tobacco consumption (yes) | 52.9 (120/227) | 50.8 (156/307) | 1.09 (0.77-1.53) | 0.639 |
| Albendazole[b] 6 months prior to study | 8.8 (20/227) | 8.8 (27/307) | 1.00 (0.55-1.84) | 0.995 |
| **Previous infections**, % (n/N) | | | | |
| Chronic hepatitis (yes) | 7.5 (17/227) | 6.2 (19/307) | 1.23 (0.62-2.42) | 0.554 |
| ITS (gonorrhea or syphilis) (yes) | 22.9 (52/227) | 18.6 (57/307) | 1.30 (0.85-1.99) | 0.219 |
| Tuberculosis (yes) | 22.5 (51/227) | 23.8 (73/307) | 0.93 (0.62-1.40) | 0.723 |
| Cough, fever or diarrhea (yes), % (n/N) | 13.7 (31/227) | 18.6 (57/307) | 0.69 (0.43-1.12) | 0.131 |
| Frequent diarrhea (>= 4 times/month), % (n/N) | 3.1 (7/227) | 4.9 (15/307) | 0.62 (0.25-1.55) | 0.300 |
| Risk group (non-heterosexual), % (n/N) | 26.9 (61/227) | 20.2 (62/307) | 1.50 (1.00-2.26) | 0.052 |
| **Current CD4+ < 200** (AIDS), % (n/N) | 9.3 (21/227) | 10.1 (31/307) | 0.93 (0.51-1.68) | 0.802 |
| **Current viral load > 2000** (detectable), % (n/N) | 14.5 (33/227) | 8.8 (27/307) | 1.76 (1.02-3.02) | **0.040** |
| **Adherence <= 95%** (bad adherence), % (n/N) | 12.8 (29/227) | 11.1 (34/307) | 1.18 (0.69-2.02) | 0.547 |
| **Hookworms in stool** | 7.0 (16/227) | 0.7 (2/307) | 10.47 (2.37-46.24) | **<0.001** |

Data shown as % (n/N) or median (IQR), unless specified otherwise. Statistically significant differences shown in bold. Percentages may not total 100 due to rounding. [a]Defined as absence of paved streets. [b]Self-medication with Albendazole (dose unknown). [c]Odds Ratio: Non-Infection is a reference category.

**Table 5. Comparison of *S. stercoralis* serology and stool examinations.**

| *S. stercoralis* ELISA | Modified Baermann Methods and/or Charcoal Culture | | |
| --- | --- | --- | --- |
| | Positive | Negative | Total |
| Positive | 75 | 86 | 161 |
| Negative | 6 | 187 | 193 |
| Total | 81 | 273 | 354 |

Sensitivity: **92.6%**; 95 IC: 0.84 - 0.97

Specificity: **68.5%**; 95% IC: 0.63 - 0.74

Positive predictive value: **46.6%**; 95% IC: 0.39 to 0.55

Negative predictive value: **96.9%**; 95% IC 0.93 to 0.99

data describing *S. stercoralis* prevalence among PLWH are scarce [8,18]. Most publications suggest a higher risk of co-infection with *S. stercoralis* in PLWH versus the general population [19–21], although PLWH do not seem to have a higher risk of disseminated strongyloidiasis, probably due to their protective Th2 cytokine patterns [22–24]. *S. stercoralis* prevalence is likely highest in impoverished tropical areas such as the Peruvian Amazon basin, although no prior studies exist of *S. stercoralis* infection in PLWH living in this region [5].

Regarding available epidemiologic data for strongyloidiasis in Latin America, a recent systematic review estimated the global pooled prevalence of HIV and *S. stercoralis* co-infection to be 5%, and as high as 8% in some Caribbean and Latin American countries (including Cuba, Bazil, and Venezuela) [22]. Studies of strongyloidiasis in Brazil, a large country that shares Amazonian areas with Peru, indicate varied prevalences in PLWH, from 2-4% in the subtropical city of Sao Paulo [25,26], to 12% in the high altitude tropical city of Minas Gerais [27] and 30% in the northeastern tropical city of Fortaleza [28]. In this last study, conducted in an area similar to ours, *S. stercoralis* prevalence was significantly higher in PLWH than in general population: 30% vs 11%. A Colombian cohort of PLWH demonstrated 0.5% stool prevalence of *S. stercoralis*, but, notably, the investigators performed only one stool test specific for *S. stercoralis* (agar culture). Ehsan et al. [18], in one of the few meta-analyses describing geographical *S. stercoralis* prevalence in PLWH, reported a pooled stool prevalence of 6.9% in Peru. To the best of our knowledge, our study is the first to provide strongyloidiasis prevalence data for one of the largest cohorts of PLWH in the Peruvian Amazon.

Expanding our review of the HIV-*S. stercoralis* coinfection literature to outside of Latin America, several publications of Asian and African PLWH utilized specific *S. stercoralis* techniques to evaluate prevalence in stool specimens. One study of nearly 1,500 PLWH from eastern India found a stool prevalence of 3.76% [29]. Similar studies describe stool prevalence of 10.8% in Laos [30], 2.4-11.5% in Ethiopia [31,32], and 8.2% in Uganda [33]. Therefore, globally, most studies of stool specimens of PLWH reveal a lower prevalence of *S. stercoralis* than our study.

Studies of *S. stercoralis* in populations without HIV living in or near Iquitos report a 10% stool prevalence among pregnant women [10], 8.7% among people living along the Nanai River [11], and 10.5% among children living in Padre Cocha [13]. Several studies published prevalences closer to ours: 16% in schoolchildren in San Martin in 1999 and 19.5% in outpatients with diarrhea in Madre de Dios in 2001 [10]. Gallardo et al. [12] reported one of the highest prevalences (28.8%), while studying stools of another vulnerable population, soldiers, in 2015. Finally, a 2005 study in a rural community of the Pasco region, which is further from Iquitos but still within the Amazon, concluded a higher prevalence than ours, 38.5% [34].

Regarding seropositivity, we found a 42.5% *Strongyloides* seroprevalence in our cohort of PLWH. Studies of other populations living in or near Iquitos also describe high seroprevalences, including 72% in a rural community 15 km from Iquitos [11], 65% in recently published cross-sectional study in general population [35], and 33% in pregnant women [10], being the latter two studies conducted by our research group, with a median difference of prevalence between stool and serum of approximately 20% [36]. Outside Latin America, Asia is the continent best represented, from which studies report seroprevalence rates ranging between 20% and 45% and indicate that seroprevalence increases with the population's degree of rurality or distance from health care [37–39].

In areas of high endemicity, *Strongyloides* crude antigen IgG ELISA is known to have a lower specificity compared to reference stool tests because (1) *Strongyloides* IgG antibodies persist for many years after treatment and (2) the crude antigen lysate allows for cross-reactivity with other helminths that can be co-endemics with *S. stercoralis* [40]. Besides, ELISA´s sensitivity may be reduced in patients with AIDS due to impaired immune responses to the parasite [41]. In our study, the specificity of the ELISA was close to 70%, similar to a study of pregnant women in Iquitos performed with the same test [10], probably due to the persistence of antibodies in an endemic area (only one patient had a positive ELISA for *S. stercoralis* with a discordant parasite in stool). About ELISA's sensitivity and negative predictive value, they were greater than 90%, higher than in the previously cited study, where it only reached 70% [10]. The higher cutoff point that we chose for this study could be responsible for this improvement.

Although various immunocompromising conditions have been associated with *Stronglyoides* hyperinfection syndrome, HTLV-1 infection (also present in Iquitos, with an estimated prevalence in general population of 1–2% [10,35] but unknown between PLWH), and iatrogenic immunosuppression via corticosteroid use are the most consistent associations [8]. PLWH, especially those with AIDS, are a vulnerable group with a relatively high utilization of corticosteroid therapy [23]. Previous investigations in PLWH [18,25,32,42] suggest that the socioeconomic status, AIDS stage, alcoholism or male gender may contribute to risk for *S. stercoralis* infection in this group [8,21].

Studies evaluating specific risk factors are rare. In our cohort, adjusted risk of *S. stercoralis* stool positivity was higher among PLWH living in a house made of wood/leaves (used as a proxy for low socioeconomic status), in a rural area (defined by the presence of unpaved streets, regardless of housing material), or with a low level of education. These findings are consistent with well-established risk factors, including poor sanitation, contact with fecal contaminated soil due to lifestyle practices, limited access to healthcare, and overall socioeconomic vulnerability [5,23,43,44]. Additionally, we found that study participants attending Loreto Regional Hospital —an urban area in the confluence of two major rivers—, had a higher risk of *S. stercoralis* infection than those attending Iquitos Hospital. Loreto Regional Hospital is the referral hospital for people from river communities with limited access to potable water, which may explain our results. This data also could be influenced by the higher amount of patients collected in this Hospital. Hookworm infection in stool was significantly associated with both *S. stercoralis* infection and seropositivity. This co-occurrence has been discussed in previous literature, as the distribution of both helminths often overlaps due to similar biological and epidemiological characteristics. Consequently, hookworm has been proposed as a proxy indicator for estimating the global burden of strongyloidiasis [11,45]. Although cross-reactivity in serology remains a concern, it did not affect ELISA´s specificity in our cohort, as previously discussed.

In our study, AIDS stage was not significantly associated with either *S. stercoralis* stool positivity or seropositivity [33]. A viral load >2000 copies/ml was significantly associated with seropositivity in the bivariate analysis, though not in the multivariate analysis. However, the substantial amount of missing data on CD4 count and viral load limits the strength of conclusions regarding the association between poor immunovirological control and infection risk. Other previously described risk factors, including a non-heterosexual sexual orientation [25,20], low educational level, agricultural occupation [31], and male sex [18,43], were borderline-associated with *S. stercoralis* seropositivity in the bivariate analysis but not found to be significant risk factors in the multivariate analysis. Deworming treatment within the past six months appeared protective against *S. stercoralis* stool positivity in the bivariable analysis but was not significant in the multivariate analysis [33]. Although albendazole (administered twice over three days) may explain this effect, many participants could not recall the exact drug dosage, which may have influenced the results.

A key strength of this study is its comprehensive approach to evaluate *S. stercoralis* infection in PLWH, utilizing two complementary classes of tests: parasitological analysis of stool samples and serological testing. Additionally, the study focuses on a population from a highly endemic Amazonian region, providing valuable epidemiological insights into an area where data on this common coinfection are scarce. Our large sample size further strengthens the reliability and generalizability of the findings.

This study has several limitations. First, we collected and analyzed only a single stool sample, which may have underestimated the true prevalence of *S. stercoralis* [46]. Analyzing three samples could have improved the detection rate of active infection [47]. To mitigate this limitation, we used two different parasitological techniques to enhance the sensitivity of larval detection. Second, the absence of molecular techniques, (e.g., PCR), which are highly sensitive and useful for identifying low-level infections, may have limited diagnostic accuracy. Additionally, our cohort primarily included PLWH who were actively engaged in routine care, possibly underrepresenting individuals not accessing care—who may be at greater risk of infection. Finally, the large amount of missing data on CD4 and viral load hindered our ability to fully explore associations between immunovirological control and infection risk. While our findings highlight a high burden of *S. stercoralis* in the Amazon, they may not be generalizable to regions with different epidemiological contexts.

## Conclusion

Infection with *Strongyloides stercoralis* is common and potentially serious among people living with PLWH in Iquitos and the surrounding areas, affecting nearly one in four individuals, primarily from impoverished backgrounds. It is essential to implement a routine screening program for *S. stercoralis*, especially at the time of entry into HIV care and during follow-up visits, due to the risk of reinfection. This measure could reduce *Strongyloides*-related morbidity and prevent severe complications, such as *Strongyloides* hyperinfection syndrome.

Furthermore, our findings underscore the urgent need for large-scale public health interventions, including deworming protocols targeted at vulnerable populations and improved access to effective screening and diagnostic tools tailored to low resource endemic regions. Proactively working toward the control of strongyloidiasis will not only improve the quality of life for PLWH but also reduce the broader public health burden in Amazonian areas.

## Supporting information

**S1 File. Graphical abstract summarizing the study protocol and its main results.**
(TIF)

## Acknowledgments

We want to thank the medical staff of the Infectious Diseases Service in Loreto Regional Hospital and Iquitos Hospital, together with the laboratory staff of LIPNAA-CIRNA and Asociación Civil Selva Amazónica for the support on the field.

## Author contributions

**Conceptualization:** Silvia Otero Rodriguez, Martin Casapia-Morales, Viviana Pinedo-Cancino, Esperanza Merino, Jose-Manuel Ramos-Rincon.

**Data curation:** Silvia Otero Rodriguez, Jose-Manuel Ramos-Rincon.

**Formal analysis:** Silvia Otero Rodriguez, Jose-Manuel Ramos-Rincon.

**Funding acquisition:** Silvia Otero Rodriguez, Jose-Manuel Ramos-Rincon.

**Investigation:** Silvia Otero Rodriguez, Seyer Mego-Campos, Victoria-Ysabel Villacorta-Pezo, Jorge Parráguez-de-la-Cruz.

**Methodology:** Silvia Otero Rodriguez, Martin Casapia-Morales, Eva H Clark, Jose-Manuel Ramos-Rincon.

**Project administration:** Silvia Otero Rodriguez, Martin Casapia-Morales, Viviana Pinedo-Cancino, Esperanza Merino, Jose-Manuel Ramos-Rincon.

**Resources:** Silvia Otero Rodriguez.

**Validation:** Silvia Otero Rodriguez, Martin Casapia-Morales, Viviana Pinedo-Cancino, Eva H Clark, Jose-Manuel Ramos-Rincon.

**Writing – original draft:** Silvia Otero Rodriguez, Jose-Manuel Ramos-Rincon.

**Writing – review & editing:** Silvia Otero Rodriguez, Martin Casapia-Morales, Viviana Pinedo-Cancino, Seyer Mego-Campos, Victoria-Ysabel Villacorta-Pezo, Jorge Parráguez-de-la-Cruz, Eva H Clark, Esperanza Merino, Jose-Manuel Ramos-Rincon.

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
