## [Decision Letter · Decision Letter 0]

High prevalence of Strongyloides stercoralis in people living with HIV: A critical health challenge in the Peruvian Amazon Basin

Dear Dr. Otero Rodriguez,

Thank you for submitting your manuscript to PLOS Neglected Tropical Diseases. After careful consideration, we feel that it has merit but does not fully meet PLOS Neglected Tropical Diseases's publication criteria as it currently stands. Therefore, we invite you to submit a revised version of the manuscript that addresses the points raised during the review process.

Please submit your revised manuscript within 60 days Jun 26 2025 11:59PM. If you will need more time than this to complete your revisions, please reply to this message or contact the journal office at plosntds@plos.org. Please include the following items when submitting your revised manuscript:

We look forward to receiving your revised manuscript.

Kind regards,

Francesca Tamarozzi

Section Editor

Shaden Kamhawi

co-Editor-in-Chief

Paul Brindley

co-Editor-in-Chief

**Journal Requirements:**

At this stage, the following Authors/Authors require contributions: Silvia Otero Rodriguez, Martin Casapia-Morales, Viviana Pinedo-Cancino, Seyer Mego-Campos, Victoria-Ysabel Villacorta-Pezo, Jorge Parráguez-de-la-Cruz, Eva H Clark, Esperanza Merino, and Jose-Manuel Ramos-Rincon. Please ensure that the full contributions of each author are acknowledged in the "Add/Edit/Remove Authors" section of our submission form.

Potential Copyright Issues:

i) Graphical Abstract. Please confirm whether you drew the images / clip-art within the figure panels by hand. If you did not draw the images, please provide (a) a link to the source of the images or icons and their license / terms of use; or (b) written permission from the copyright holder to publish the images or icons under our CC BY 4.0 license. Alternatively, you may replace the images with open source alternatives. See these open source resources you may use to replace images / clip-art:

ii) Figure 1. Please (a) provide a direct link to the base layer of the map (i.e., the country or region border shape) and ensure this is also included in the figure legend; and (b) provide a link to the terms of use / license information for the base layer image or shapefile. We cannot publish proprietary or copyrighted maps (e.g. Google Maps, Mapquest) and the terms of use for your map base layer must be compatible with our CC BY 4.0 license.

1) State what role the funders took in the study. If the funders had no role in your study, please state: "The funders had no role in study design, data collection and analysis, decision to publish, or preparation of the manuscript.".

**Reviewers' Comments:**

**Comments to the Authors:**

**Please note that one of the reviews is uploaded as an attachment.**

Reviewer's Responses to Questions

**Key Review Criteria Required for Acceptance?**

**Methods**

-Are the objectives of the study clearly articulated with a clear testable hypothesis stated?

-Is the study design appropriate to address the stated objectives?

-Is the population clearly described and appropriate for the hypothesis being tested?

-Is the sample size sufficient to ensure adequate power to address the hypothesis being tested?

-Were correct statistical analysis used to support conclusions?

-Are there concerns about ethical or regulatory requirements being met?

Reviewer #1: -Are the objectives of the study clearly articulated with a clear testable hypothesis stated? YES

-Is the study design appropriate to address the stated objectives? THE DESCRIPTION OF STUDY DESIGN SHOULD BE REVISED

-Is the population clearly described and appropriate for the hypothesis being tested? YES

-Is the sample size sufficient to ensure adequate power to address the hypothesis being tested? YES

-Were correct statistical analysis used to support conclusions? YES

-Are there concerns about ethical or regulatory requirements being met? YES

Reviewer #2: The Methods are very well laid out and other workers can repeat these studies. The population size is adequate and well described. The statitstical methods are sound and adequate. There are no ethical concerns.

**Results**

-Does the analysis presented match the analysis plan?

-Are the results clearly and completely presented?

-Are the figures (Tables, Images) of sufficient quality for clarity?

Reviewer #1: YES

Reviewer #2: The analysis matches the the plan except where no results are given for the direction examination method which was used. the authors should mention this even if no parasites were found using the method. The table and figures are well presented.

**Conclusions**

-Are the conclusions supported by the data presented?

-Are the limitations of analysis clearly described?

-Do the authors discuss how these data can be helpful to advance our understanding of the topic under study?

-Is public health relevance addressed?

Reviewer #1: -Are the conclusions supported by the data presented? YES

-Are the limitations of analysis clearly described? YES

-Do the authors discuss how these data can be helpful to advance our understanding of the topic under study? I think the data should be discussed in more detail in some important issues

-Is public health relevance addressed? YES

Reviewer #2: The Conclusions are supported the findings of the study. The limitations and strengths of the study are well laid out. the authors also showed how the study could exapand public health efforts. Again, this could be strengthened if data were available on other helminth infections which could be incorporated in control measures.

**Editorial and Data Presentation Modifications?**

Reviewer #1: (No Response)

Reviewer #2: Abstract

1. Change “prevalent” to important

2. State “sociodemographic” factors instead of just factors

3. Change “via” to “using”

Line 112. Strongyloidiasis is also caused by Strongyloides fuelleborni. Authors should state that it’s caused by S. stercoralis in the study region.

Line 139. Delete “clinically”

Line 142. Change “tests” to “test”

Line 164. Authors should state why a single stool sample was taken instead of two or three which would improve sensitivity of parasitological diagnosis

Line 166. Change via to using

Line 178-179. Would centrifugation of the sediment improve detection rate?

Line 181. Change “is” to “was”

Line 182-184. How was charcoal culture examined?? Stereomicroscope? Compound microscope??

Line 187. Change RPM to relative centrifugal force

Line 201. Change “presents” to “presented”

Line 206. Strike “parasitological”

Line 216. What about the result

Line 290. “Though” should be “although”

Line 293. “Though” should be “although”

Line 358-371. The authors should focus their discussion on the multivariate analysis. Also point of the study participants had very low viral loads.

**Summary and General Comments**

Reviewer #1: Dear authors, this paper addresses an important issue, not only because of the coinfection HIV- S stercoralis, but also because it provides data about strongyloidiasis prevalence. However, I think the paper can be improved if you accept the next comments, if worthy and appropriate for you.

• Line 85 (abstract): “…as the parasitological gold standard…” As there is no gold standard diagnosis for S. stercoralis, I will advise to change by “more sensitive” parasitological method , to avoid confusion of concepts

• Line 117: “Strongyloidiasis is clinically important because…” I think the expression clinically is not correct, as strongyloidiasis is pauci-symptomatic. Maybe, epidemiologically, or just to say important.

• Line 133-134: “To the best of our knowledge, no published parasitological or Sero epidemiological studies exist on S. stercoralis infection in PLWH in Peru.”

See García C, Rodríguez E, Do N. López de Castilla D, Terashima a, Gotuzzo E. intestinal parasitosis in patients with HIV-AIDS. Rev Gastroenterol Peru. 2006;26(1):21–24

• Line 139: “…is not clinically available…” better to say …is not available

• Lines 142-143: “Overall, no available diagnostic tests can serve as a gold standard for the diagnosis of strongyloidiasis” I would advise to precise that a combination of techniques is the approach which allows a higher sensitivity (see Diagnostic methods for the control of strongyloidiasis, Virtual meeting, 29 September 2020. Geneva: World Health Organization; 2020.)

• Line 150: “…prospective, cross-sectional study …” The study design cannot be both prospective and cross-sectional!

• Line 183: “…to observe adult larvae before…” Do you mean filariform larvae?? Or adult???

• Line 193: “Considering the anticipated high seroprevalence and…” as you are including outpatients and inpatients and the last are the most representative in the sample, it is difficult to understand why you anticipated high seroprevalence ¿?

• Lines 279-280: “ WHO (…) emphasize the need for clear guidelines on whether preventive treatment for strongyloidiasis should be recommended,…” Actually, the reference you include (15) is the WHO guidelines, even though the guideline accepts the need of more evidence for strong recommendation

Major comments

• In the discussion, ELISA’s specificity and negative predictive values are assumed as high in the sample. However, severely immunosuppressed patients will have impaired their immunological response. The fact that serology can be inconclusive has been highlighted in these patients. (Nabha L, Krishnan S, Ramanathan R, et al. Prevalence of Strongyloides stercoralis in an urban US AIDS cohort. Pathog Glob Health. 2012;106(4):238-244. doi:10.1179/2047773212Y.0000000031). In the discussion you mention (line 359) that “…In our study, AIDS stage was not significantly associated with S. stercoralis stool positivity or seropositivity. However, the Current viral load > 2000 was significantly associated with S. stercoralis serological status (table 4). I think it is important to include and discuss this issue. In this line, I would advise you to reconsider also the sentence in the abstract, about the PNV, Line 98, “…..which is useful for ruling out the presence of active infection” ¿?.

Also, direct observation was used in the analysis of the samples, but there is no mention of other STH or parasites. However, in view of the incidence of diarrhoea in the sample (table 1) and considering that the S. stercoralis infection is pauci-symptomatic, it would be important to mention that other STH, e.g., hookworm, can be co-infecting the patients, and the relationship with the positive serology in those patients, that is, the impact of this fact in the sensitivity of the test.

• Lines 92-95 (abstract)Lines 254-255: “risk factors for stool positivity included living in a rural area (adjusted OR: 2.16), whereas significant risk factors for both stool and seropositivity included living in a house made of wood/leaves (adjusted ORs: 2.10 and 2.48, respectively) or in the Loreto Regional Hospital catchment area (adjusted ORs: 5.66 and 5.37, respectively). Is the Loreto Regional Hospital catchment area an urban area? Are houses in rural areas made of different materials than wood/leaves? I mean, are those risk factors logical or are they confusing factors? To explain more in detail in the discussion

• 368-371: “ treatment in the previous 6 months was a protective factor for S. stercoralis stool positivity in the bivariable analysis but not in the multivariate analysis [32], probably because ivermectin (the treatment of choice for strongyloidiasis) use was uncommonly reported by our study participants…” I do not understand what this means exactly. Maybe you can rephrase it. Also, Albendazole can be effective for strongyloidiasis, if administered twice in 3 days. Maybe you will have this information ¿?

• Line 345: “…have hypothesized … ” Hypothesized??

• Lines 374-375: “This dual methodology enhances diagnostic accuracy by addressing both

active infections and past or subclinical exposure” I think this sentence is not correct. First, it is well known that the strongyloidiasis diagnosis benefits from a combination of techniques. Second, you cannot differentiate between active infections and past or subclinical exposure in a population living conditions you have describe as poor, related to hygiene and sanitation, where re-infection will be common.

Lines 383-385: “….as PCR has a higher sensitivity than either of the parasitological methods we employed and could identify low-level infections” This is not correct; even though the PCR is slightly more sensitive that Baermann technique in most of the studies, the test is negative in a high proportion of Baermann positives (see. Epidemiology of intestinal helminthiases in a rural community of Ethiopia: Is it time to expand control programs to include Strongyloides stercoralis and the entire community? PLoS neglected tropical diseases, 14(6), e0008315. https://doi.org/10.1371/journal.pntd.0008315)

Reviewer #2: This is a timely study of co-infection with two serious pathogens one of which is a highly understudied and litte understood parasite S. stercoralis. The paper can be imporved by paying attention to the following;

Introduction

The Introduction has not made the case for studying the epidemiology of S. stercoralis in PLWH as there is no deleterious outcome of either the virus or parasite from co-infection. However, the study is important as S. stercoralis which is treatable, can cause longstanding infections and can have complicated and fatal outcome. Diagnosing and treating S. stercoralis in PLWH remove a possible cause of severe morbidity and mortality from a vulnerable population.

Methods

These are well laid out. However, the authors mentioned that they used direct stool examination after staining with Lugol’s iodine but show no results from this method. If no parasites were detected the authors should say this as this is a Result.

Results

The authors did not mention finding any other helminths using the methods at hand. This is important while discussing the matter of cross-reactivity of the ELISA. If none were found this should be stated.

Discussion

First paragraph should be used to discuss the major finding of the study.

The presence or absence of other helminths should be stated in the study. If none were found then other references should be cited on this to both account for cross reactivity and the approach to control of soil transmitted helminths in the area including S. stercoralis.

PLOS authors have the option to publish the peer review history of their article (what does this mean? ). If published, this will include your full peer review and any attached files.

**Do you want your identity to be public for this peer review?** For information about this choice, including consent withdrawal, please see our Privacy Policy .

Reviewer #1: No

Reviewer #2: No

**Figure resubmission:**

**Reproducibility:**



---

## [Editor Report · Decision Letter 1]

Dear Dr. Otero Rodriguez,

Thank you for submitting your manuscript to PLOS Neglected Tropical Diseases. After careful consideration, we feel that it has merit but does not fully meet PLOS Neglected Tropical Diseases's publication criteria as it currently stands. Therefore, we invite you to submit a revised version of the manuscript that addresses the points raised during the review process.

Please submit your revised manuscript within 30 days Jul 04 2025 11:59PM. If you will need more time than this to complete your revisions, please reply to this message or contact the journal office at plosntds@plos.org. Please include the following items when submitting your revised manuscript:

Response to Reviewers Revised Manuscript with Track Changes Manuscript

Shaden Kamhawi

co-Editor-in-Chief

Paul Brindley

co-Editor-in-Chief

**Additional Editor Comments:**

Please, revise accurately the newly added paragraphs - some sentences are not clear and need grammar editing.

Other comments:

Introduction (lines 135-138) - I suggest revising this paragraph as follows: "The World Health Organization (WHO) includes Strongyloides among the soil transmitted helminthiases targeted for improved control by 2030; for this purpose, specific guidelines have been published recently. However, the WHO [...]"

Methods, paragraph 2.3: I would delete the first sentence ("We established...")

Please, when refering to larvae avoid the term "adult", which generates confusion and is not appropriate; use either lavae or adult worms.

**Journal Requirements:**

1) We do not publish any copyright or trademark symbols that usually accompany proprietary names, eg ©,  ®, or TM  (e.g. next to drug or reagent names). Therefore please remove all instances of trademark/copyright symbols throughout the text, including:

- © on page: 28.

2) Please ensure that the funders and grant numbers match between the Financial Disclosure field and the Funding Information tab in your submission form. Note that the funders must be provided in the same order in both places as well. Currently, the order of the grants is different in both places.

**Figure resubmission:****Reproducibility:** To enhance the reproducibility of your results, we recommend that authors of applicable studies deposit laboratory protocols in protocols.io, where a protocol can be assigned its own identifier (DOI) such that it can be cited independently in the future. Additionally, PLOS ONE offers an option to publish peer-reviewed clinical study protocols. Read more information on sharing protocols at https://plos.org/protocols?utm_medium=editorial-email&utm_source=authorletters&utm_campaign=protocols

---

## [Editor Report · Decision Letter 2]

Dear S Otero-Rodriguez,

We are pleased to inform you that your manuscript 'High prevalence of Strongyloides stercoralis in people living with HIV: A critical health challenge in the Peruvian Amazon Basin' has been provisionally accepted for publication in PLOS Neglected Tropical Diseases.

Best regards,

Dora Buonfrate, M.D., D.T.M.&H., PhD

Guest Editor

Francesca Tamarozzi

Section Editor

Shaden Kamhawi

co-Editor-in-Chief

Paul Brindley

co-Editor-in-Chief

---

## [Editor Report · Acceptance letter]

Dear Ms Otero Rodriguez,

We are delighted to inform you that your manuscript, "High prevalence of Strongyloides stercoralis in people living with HIV: A critical health challenge in the Peruvian Amazon Basin," has been formally accepted for publication in PLOS Neglected Tropical Diseases.

Best regards,

Shaden Kamhawi

co-Editor-in-Chief

Paul Brindley

co-Editor-in-Chief
